# Regulating Gambling Use through the Overton Window: From an Addictive Behavior to a Social and Epidemiological Problem

**DOI:** 10.3390/ijerph20085481

**Published:** 2023-04-12

**Authors:** Antonio Jesús Molina-Fernández, Anna Robert-Segarra, José Antonio Martín-Herrero, Iván Sánchez-Iglesias, Jesús Saiz-Galdós, Karla Fernández-Mora

**Affiliations:** 1Department of Social, Work and Differential Psychology, Complutense University of Madrid, 28223 Madrid, Spain; 2Benito Menni Centre de Salut Mental d’Adults (CASM), 08830 Sant Boi de Llobregat, Spain; 3Department of Social Psychology and Anthropology, University of Salamanca, 37005 Salamanca, Spainidu003840@usal.es (K.F.-M.); 4Department of Psychobiology & Behavioral Sciences Methods, Complutense University of Madrid, 28223 Madrid, Spain

**Keywords:** gambling regulation/legalization, recreational use, qualitative study, Overton window

## Abstract

During the last decade, gambling (online and offline) regulation has become a social and epidemiological problem all around Europe. The aftermaths of this addiction have increased since the so-called “responsible gambling law”, in the second decade of the 21st century. The Overton window (OW) strategy is a political theory that describes how the perception of public opinion can be modified so that ideas that are inconceivable for society become accepted over time. The objective of this study is to identify whether an OW has been used to bias the adequacy of the gambling debate, as well as its scientific, legal, and political bases and the main consequences for both the general population and the major risk groups, especially the consequences in social and health contexts. The study was conducted by the application of the historical-logical method as the central axis of analysis and reflection, and the technique of qualitative research content analysis as a procedure in the process of execution of the scientific task, related to a historical trend study of the research object. The main consequences found were: the political acceptance of gambling for economical causes and taxes benefits, the use of popular characters to increase the acceptance of the pattern of behavior, the inclusion of the gambling operators as agents in the risks control, and the absence of intervention until the main consequences have been transformed into an epidemiological problem (with social aftermaths higher than the previously identified related to the gambling problems). Furthermore, the results suggest the need to implement prevention and health promotion strategies and the adoption of specific legal measures that regulate the access and the marketing of gambling operators’ activities.

## 1. Introduction

The appearance in 2011, in Spain, of the so-called “Responsible Gambling Law” and its liberalization of advertising and access to all types of gambling through both online platforms and face-to-face services have completely changed the perception and social impact of gaming and gambling. The freedom of access to the great waves of content and services, including games, online betting, lotteries, etc., and the profound transformation that communication is undergoing thanks to the interactivity of some supports, such as chats, forums or e-mail, have generated a series of transformations and risks, such as: exposure to stimuli, the linking of said stimuli to figures benchmarks in various fields (basically, athletes, and artists), the increase in exposure time to these screens, and the compulsive use of this technology [1,2]. According to the report “European School Survey Project on Alcohol and Other Drugs” (ESPAD), “The high degree of normalization of gambling in societies and the culture of gambling within the family environment have been recognized as important drivers of gambling onset and youth progression into problem gambling”. The 2019 ESPAD results show that gambling for money has become a popular activity among school students in Europe, with 22% of respondents reporting gambling on at least one game in the past 12 months (predominantly lotteries). An estimated 7.9% of students had gambled for money online in that period, probably without the consent of their parents or caregivers. The screening tool used in the latest survey to estimate problem gambling revealed that, on average, 5% of students who had gambled in the last 12 months fell into this category [3].

Fenichel found essential elements common to all addictive disorders, regardless of whether they are addictions with substances or behavioral addictions, such as the initial pleasure mechanisms, the subsequent habituation mechanisms and the low perception of the problem that people present, including the denial of addiction and the phenomenon of attribution/external locus of control [4,5]. Therefore, there have been and are various factors that influence the development of problematic gambling behavior and gambling addiction until it becomes a pathology with multiple causes and consequences, having to be considered as a problem with no linear etiology, multifactorial, and multi-causal.

In this way, it is verified that a consensus on “addictions without substance” has been developed relatively recently, specifically on pathological gambling and gaming disorder, which has been changing since 1980 [6]. It was in the DSM-5 [6] that pathological gambling (such as gaming disorder) was included in the category of “addictive disorders” together with “addictions with substances” and with the name of “pathological gambling”, defined as “inability to reduce or eliminate excessive participation in gambling involving monetary bets, despite the severity of its negative consequences” [7,8]. Within the DSM-5, it becomes evident that, to diagnose “pathological gambling”, it must be a type of gambling that is “problematic, persistent and recurrent, and must cause clinically significant deterioration or discomfort” [7,8].

The relevance of gambling all around the world is increasing since the regulations of online gambling during the 21st century: there have been clear variations since 2000 in gambling disorder rates across different countries in the world (0.12–5.8%) and in Europe (0.12–3.4%) [9]. The influence of the policies in these is increasing and the strategies for their implementation are relevant topics for the understanding of the phenomenon [9].

Overton window (OW) is a political theory that describes how the perception of public opinion can be modified so that ideas that are inconceivable for society become accepted over time [10]. The window metaphor is used to describe a well-defined space, from which inconceivable ideas can be considered viable in a social context once they have been declared and defended, framed and defended again [10]. It is about exposing an idea that, despite the fact that it initially seems unacceptable, can be defended and framed so that little by little it becomes an understandable idea for society [11]. As a political strategy, it is a maneuver intentionally and strategically built by someone in order to modify society’s appreciation of a taboo topic to later introduce changes in the political and social framework [12].

### 1.1. First Stage: From the Unthinkable to the Radical

In this first stage, the approval of the fact that one wishes to introduce is still something unthinkable. The practice is at the lowest level of acceptance of the OW of possibilities (still very narrow, not to say tightly closed), since society considers this an action alien to public morality. To change the perception that society has about the subject, it is necessary to transfer the question to the scientific sphere. In the same way, groups or organizations can be created to support the idea to be implemented. Once the transition from the original negative attitude to a more open and positive attitude has been given in society, reinforced with the help of socially accepted groups, the objective of the first phase is considered achieved: the perception of initial rejection is questioned, the originally unacceptable topic begins to be discussed openly.

### 1.2. Second Stage: From the Radical to the Acceptable

In this second stage, the social approval of the fact is frankly pursued, at first inconceivable. For this to be accepted, it is necessary that the opinions of “scientists” continue to be disseminated to promote a favorable discourse. In the same way, it is necessary to insist on how opportune it is not to have prejudices on the subject, describing as intransigent those who refuse to share their ideas about it. In order to eliminate the negative connotations attributed to the original term, it is necessary to create a euphemism; separate the word from its meaning. To reinforce the legitimacy of the activity to be preached, at the same time a historical, mythological, or invented precedent is created, which serves as a reference and can be used as proof of the authenticity of the action. The combined use by the media and pressure groups will end up making the fact acceptable, sooner rather than later.

### 1.3. Third Stage: From the Acceptable to the Sensible

At this point, it seeks to make sensible what at first was totally unacceptable. In order to root the intended objective in the social fabric, an effort is made intentionally to convert ordinary people who show ideas contrary to the consolidation of the new belief into radical enemies. During this same stage, the media and the scientific community try to convince society about the naturalness and normativity of the idea that is being propagated. With the actions developed, the final objective of this third stage is that the idea to be implemented is considered a reasonable custom.

### 1.4. Fourth Stage: From the Sensible to the Popular

At this moment, the media, supported by famous people and authorities, speak openly about the matter in question, thus becoming a more present element in a multitude of phenomena of social life. The belief that is intended to establish begins to appear in movies, lyrics of famous songs, videos, or flooding social networks in all kinds of formats. The phenomenon achieves international attention and becomes unstoppable and massive. In addition, to reinforce its positive image, use is made of normative individualism to justify the acceptance of the behavior.

### 1.5. Fifth Stage: From the Popular to the Political

The last phase involves starting to prepare legislation to regulate the new phenomenon. Interested groups consolidate their power and publish polls that supposedly confirm a high percentage of supporters in favor of the recently approved measures. During the last stage of the OW movement, from the popular to the political, with the approval of new laws, society can be fractured due to divergences of opinion. However, if the political process has been carried out successfully, the number of followers who will defend the new belief will be greater than the minority group that does not approve.

The movement of the windows is a perfectly defined strategy to transform the social gaze of a topic that was initially inconceivable [10].

### 1.6. Objective

The objective of this study is to identify whether an OW strategy (a political theory that describes how the perception of public opinion can be modified so that ideas that are inconceivable for society become accepted over time) has been used to bias the adequacy of the gambling debate, as well as its scientific, legal, and political bases and the main consequences for both the general population and the major risk groups, especially the consequences in social and health contexts.

## 2. Materials and Methods

### 2.1. Type of Analysis

This study draws on qualitative research methodology [13], specifically on historical research [14]. Historical research has a scientific character, because in order to know its object of study, it is done through rules and procedures of the scientific method. It is a humanistic discipline since historians analyze and record the individual and group events in society, the role that individuals perform in institutions and, fundamentally, the meaning of events in the context where they arose [15]. Aróstegui points out that it must contemplate a theme, a project and a procedure to address it. The historian must have a design that guides his work and guides him towards the search for the respective conclusions. To plan a historical investigation should take into account the technical and cognitive moments, in addition to attending to the levels of: that of what one wants to know, that of how to know, and that of the checking the known [16].

The main framework has been the hermeneutic method: the term hermeneutic comes from the Greek word *hermeneuo*, which means “I explain”. This method proposes that all understanding is different; its meaning is determined by the historical situation of the interpreter and for the interests of each era in its purpose to understand itself, in the light of tradition. In historical research, objectivity can only be reached from a certain distance that allows a broader vision and a full recognition of the fact under study. The distance in time allows the researcher to take awareness regarding prejudices and discriminate what is false from what is true. The understanding of the past must be done from the present, although this in turn is the result of the past [16].

Our aim was to examine the consequences at all levels of the legalization and regulation of gambling in Spain since the approval of the responsible gambling law in 2011. In order to clearly specify the object of study, a series of categories (the levels of the OW) were identified a priori. In this way, the suitability of the categories chosen could be certified [17]. We expect that OW’s model could give us the framework to explain the quick and efficient evolution of a phenomenon from a consideration of individual/behavior problem to a collective/social problem, integrated in our actual social context without the chance to be modified, erased, or decreased in its influence [10].

### 2.2. Information Search

The existing literature review has been conducted, to the extent possible, following the PRISMA guidelines for systematic reviews [18]. In this way, we intended to have an unbiased sample of records that addressed the subject of our study.

#### 2.2.1. Eligibility Criteria, Information Source, and Search Strategy

To be included in this review, the studies had to be published online between 2011 and 2022, both inclusive, in Spanish or English.

Data were extracted from websites and reports about gambling, entering the following Boolean expression in the Google search engine: ((“gambling*” [Title/Abstract] OR “juego patológico*” [Title/Abstract] OR “gambling addiction” [Title/Abstract]) AND (“adicción a juego” [Title/Abstract] OR “trastornos adictivos” [Title/Abstract] OR apuestas [Title/Abstract] OR bets [Title/Abstract] OR gambling addiction [Title/Abstract] OR “ behavioural addiction” [Title/Abstract] OR “gambling use” [Title/Abstract])) NOT (drugs [Title] OR “substance” [Title] OR “substance treatment” [Title] OR “drugs treatment” [Title]).

#### 2.2.2. Selection and Data Collection Process

Researcher A.J.M. and researcher A.R., on their own, retrieved all eligible records, and individually read these reports to determine their final inclusion.

#### 2.2.3. Data Items Analysis

The information collected was analyzed through Atlas.ti 8 software, a digital tool that allows the import of documents and the systematization of relevant data. Coding process was discussed by the study team, in order to reduce dissimilarities and maintain an inter-coder reliability suitable to respond to the objectives of the study. At last, the information provided in the previous phases was compared according to the categories established in the first instance [13]. Researcher A.J.M. and researcher A.R., on their own, searched for and extracted the contents for each phase of the OW. The researchers also looked for interpretations of the significant findings, in both the results and discussion section of the paper. Cases of disagreements were settled by consensus and with the aid of a third researcher (J.A.M).

## 3. Results

Figure 1 summarizes the process of identification and selection of reports in this review.

Below, the main information and results are presented by stages of the Overton’s window model.

### 3.1. First Stage: From the Unthinkable to the Radical

As a historical background of pathological gambling, we can include the regulations that were made on games of chance in Antiquity, finding restrictive laws on games of chance in both the Egyptian Empire and Rome or ancient Greece, due to the danger that these civilizations detected in gambling as dice or playing cards. In the twelfth century, in the Middle Ages, games were prohibited by the Church in Europe on several occasions, due to their sinful and stultifying condition of men, leading Saint Thomas Aquinas to classify chance and its dangers. In any case, as has happened in a general way with the game, this message had (and has) an ambivalence component, since at the same time the church itself was financed with lotteries and various raffles, as happens nowadays with national lotteries.

Regarding the classification of what types of games are problematic and dangerous, it changed during these times. The first document where a medical approach is used to analyze gambling as a mental problem is found in 1561, by Pascasio Justo, who used the theory of humors to treat gambling problems. This approach, of the humors and temperaments, very consistent with the science and morals of the time, remained stable until the appearance at the end of the 19th century of the “Morel’s Theory of degeneration”, within primitive psychiatry, in which it used the argument of “mental weakness” to justify the propensity of some people to suffer more from this and other types of disorders. From the 19th century, it is also one of the best and most complete descriptions of the pathology of compulsive gambling ever written, and it does not correspond to the academic or scientific publication, but to the novel: in “The Gambler”, where Fyodor Dostoevsky describes with great precision the stages of development and relapse of a gambler, as well as attempts to remit their addiction. In this case, he explains a case of addiction to gambling (specifically, roulette), with a precise description of the traits and emotions of the player who gives the novel its name.

Before the rise of the Internet, access to gambling was very constrained, which was a strength. Since the very first European settlers came, there has been a cyclical link between gambling and the United States. Those moving from England had a more tolerant view of gambling and were more than glad to allow it, in contrast to Puritan bands of settlers who openly forbade it in their new communities. Public outcry resulted in a national ban on gambling in 1910. Similar to the alcohol ban of the time, it was fairly challenging to police, therefore gambling persisted with only a minimal amount of secrecy. Gambling was a result of the Wall Street Crash and the Great Depression that it caused in the early 1930s. The situation led to gambling being legalized again, as for many, this was the only prospect of alleviating the grinding poverty which they suffered through. Somehow seeking they wanted to recover the losses caused by the financial “crack”.

### 3.2. Second Stage: From the Radical to the Acceptable

A paradigm that examined addictions without substances was not developed until after World War II. Fenichel was the first author to create a taxonomy with a scientific foundation that covered addictions without drug abuse. This classification remained mostly consistent until the 1980 addition of pathological gambling to the DSM-III Diagnostic Manual [19]. Pathological gambling was identified as a cognitive disorder that affects slightly more than 1% of the population by medical authorities in several countries during the 1980s and 1990s, and numerous treatment and therapy programs (basically recovery and the 12 steps) were created to address the issue. All those years have seen the continuation of this contradictory relationship. The decision to enter the world of online casinos was made back in 1994. When the Internet started working, casinos too started operating online. Poker, slots, and bingo were a few of the real money games that were accessible to online gamblers. Online registration for games increased as they got better in terms of sound, graphics, and overall functionality.

### 3.3. Third Stage: From the Acceptable to the Sensible

Since online casinos have developed, everyone can bet whenever and wherever they choose. In Western countries, four out of five people gambled at least occasionally at the start of the twenty-first century. The 20th century saw an increase in gambling, which brought attention to the compulsive gambling problem, which affects people’s ability to regulate or limit their gambling. There has been a surge in interest in Internet gambling ever since many nations have legalized it (as happened in Spain in 2011). The world has witnessed efforts to legalize it on a country-by-country basis as well as the explosive growth of mobile gaming. The best online casinos have identified a market and have stepped up to deliver. More than this, immediately gambling operators linked their images to worldwide famous sportsmen and clubs, such as Real Madrid, Lionel Messi, Cristiano Ronaldo, Rafael Nadal, Neymar Jr… at the same time as other celebrities and stars were recognized as users of gambling online.

### 3.4. Fourth Stage: From the Sensible to the Popular

It is safe to assume that desktops are falling far behind in favor of more mobile alternatives as a wave of spectacular mobile-focused online gambling sites sweeps the globe. It has brought gambling closer than ever because the majority of online gambling platforms now allow connectivity with mobile devices. With this growth, it appears that cryptocurrency, easy credit, readily available credit cards, debt consolidation, low-interest personal loans, and gaming are the next step in the evolution of gambling. There are now online gaming sites where you may bet using digital currencies. In a nutshell, technology has made a significant contribution to the global evolution of gaming. With the integration of cryptocurrency into the gambling industry, players can now withdraw and deposit funds without using a third-party financial institution. Anyone can make deposits and withdrawals at online casinos using anonymity without the other person finding out what they are doing. The COVID-19 epidemic has altered people’s online usage patterns and connection habits. A rise in socially acceptable gambling behavior may be correlated with feelings of anxiety, loneliness, stimulation demands, and reinforcement. Online treatment solutions were once again discussed during the epidemic, when receiving in-person treatment for gambling disorders became impossible. Internet access is a double-edged weapon once more. Open the door to your gambling issues on one side, and the window to your rehabilitation on the other.

### 3.5. Fifth Stage: From the Popular to the Political

Hence, the future of gambling is lying directly in the hands of technology and the work of professionals in the design of attractive platforms, psychologists who manage the motivation of the possible target “victim”. This is called persuasive design, the application of psychology to design more effective, more fun, and more engaging apps and devices. Creating persuasive technologies is probably one of the best ways of improvement for business plans, enterprises’ economic and financial benefits. These are the reasons why it is just about as difficult to predict the future for gambling as it is to uncover some of the origins of the gambling games we know so well today. The focus is currently on the mobile gaming industry, where online bookmakers offer a wide variety of wagers and games that are simpler to use, quicker to access, and give users the opportunity to engage with real people in real time. The use of virtual reality technology is just beginning to be practical, and it is likely that the next phase of gambling will be related to VR: the opportunity to determine as a social activity with your own “friends,” in a real “room” shared with other people, and to receive all the stimulations without the risk and consequences of leaving home is really alluring and beneficial for gambler operators and involved stakeholders. How about playing poker online in the comfort of your house with a large group of your friends from across the world while you have a few laughs and try to identify telltale facial ticks? In a very short period of time, it would be promoted as “The new reality of fun and leisure time”, heavily based on individual choices and supported by the murky areas of actual policies and regulations. Right now, it is probably being tested out (looking for the best balance between stimulants and behavioral responses).

That is the reason why several international organizations, such as Pompidou Group of the European Commission, are creating and developing forums to implement policies about decreasing the offer and increasing the responses to the problems related to online gambling and its derivates.

Figure 2 summarizes the OW phases and the results on gambling use and regulation, while Table 1 illustrates the steps with textual citations.

## 4. Discussion

The main consequences found were: the political acceptance of gambling for economical causes and taxes benefits, the use of popular characters to increase the acceptance of the pattern of behavior (20), the inclusion of the gambling operators as agents in the risks control, and the absence of intervention until the main consequences have been transformed into an epidemiological problem, with social aftermaths higher than the previously identified related to the gambling problems [9,19]. Furthermore, the results suggest the need to implement prevention and health promotion strategies and the adoption of specific legal measures that regulate the access and the marketing of gambling operator’s activities [9].

In this way, the Spanish Law 13/2011, of May 27, regulating gambling, mentions cross-cutting actions of various types—preventive, awareness-raising, intervention, and control. The purpose of these is to achieve good gaming practices, promote moderate and non-compulsive gaming attitudes, prevent possible effects that an excessive practice can produce, and protect minors and other groups at risk. In addition, within the framework of corporate social responsibility, responsible gambling policies also require gambling operators to establish basic rules and draw up a plan of measures to minimize the possible harmful effects derived from gambling. Subsequently, this law is specified in a decree (958/2020) that specifies issues related to commercial communications (including different ethical principles of mandatory compliance), responsible gaming policies, and the protection of consumers. For this protection, the implementation of control systems is needed to detect possible risky behavior, as well as provisions on the suspension of gaming accounts due to self-exclusion and self-ban.

This type of regulation seeks a balance between, on the one hand, the freedom of individuals to bet and invest their money, and, on the other hand, to protect the population from possible abuses exercised by the operators when designing, advertising, or allowing gambling. These regulations seek to avoid conflicts, and from a controlled permissiveness, to allow the freedom of betting, where (gaming and) gambling is allowed, but imposing assumptions. They make the user responsible, without completely exempting the operator from responsibility, but neither without penalizing him for the possible development of addictive behaviors.

As a counterpoint to people’s freedom to play, is the freedom to seek help online. Thus, there are self-help resources, apps to monitor self-control of addictive behaviors, the possibility of limits on screen time, and the opportunity of accessing mutual aid groups or a therapist directly. From public health, it seems essential to have web pages and resources well positioned in search engines to offer help, as well as to monitor and follow up on prevention and intervention programs, to assess their effectiveness in different population groups and their possibility of generalization.

Gambling behavior can be explained on the basis of the OW [10]. As a behavior subject to social, economic, political, and even medical conditions, gambling requires a phased analysis such as the one proposed. In this analysis, the debate about the legitimacy of gambling is at crucial point. The use has evolved from the 1970s until the present moment, in which positions based on individual rights and freedoms allow us to speak openly about legalization and/or regulation of the different uses. In another order of topics, in relation to the “labeling” of gambling behavior, “gambling disorder” is included within the DSM-5 [7]; in the chapter “Substance-related and addictive disorders”. Section III includes “Internet Gaming Disorder” [7]. The difference between the two is that “gaming disorder” only includes Internet games without betting, while recreational or social activities, professional networks, and Internet pornography are also outside this category [20]. When playing for money on the Internet, the DSM-5 calls it “gambling”. In this way, “Gaming Disorder” has received greater recognition than other behavioral addictions, such as sex addiction or shopping addiction, which have not been included in the DSM-5 because there is not enough evidence about them [21]. Even within the DSM-5, the possibility of including addiction to other types of games is included, even if they do not require an Internet connection or the purpose is to create a parallel virtual reality [20]. In another field of consideration, pathological gambling is labeled as “deviant behavior” [21] in so far as it transgresses the rules established by the prevailing social groups. To the extent that gambling loses its symbolic function (to link with the other), it becomes only a vehicle for gain (or, rather, for economic loss) in the loneliness of self-satisfaction. When motivation is high (to win money), accessibility is good (from a mobile phone 24/7) and the craving is up, gambling disorders are guaranteed.

Gambling addiction can mean the comorbid presence of other mental health problems, for example: a depressive disorder, an autism spectrum disorder, an anxiety disorder, or even a post-traumatic stress disorder, among others. Beyond diagnosing dysfunctional behavior in relation to gambling, the possibility of making a functional diagnosis should be considered, that is, to consider what is the function of the addictive behavior for the person on its totality. Only by encompassing the individual as a whole (behavioral-rational-emotional and sensory) and in their interaction contexts, can problem behavior be understood and treated.

The fast evolution of gambling, and its connection with gaming, especially when players are involved in massively multiplayer online role-playing games (MMORPGs), can be a way to erase the differences between “gaming” and “gambling”, at less the topic related with the use of money. The use of loot boxes in videogames as a way to increase the investments and the benefits can be the new problem for the next generation about bet addiction [22]. We also came up with an educational approach to the population, especially adolescents, as a preventive measure, that tries to warn them of the dangers of pathological gambling, of how easy it is for sporadic use of gambling to become a maladaptive pattern, and that we should be able to teach our teenagers other ways to escape, to get pleasure, or simply to interact [23].

We propose an approach that would mix the basic concepts of addiction (cravings, loss of behavior control, withdrawal symptoms, unsuccessful attempts to control and quit addictive behavior, frequent relapses that alter the continuity of your personal, family, social, economic, legal life, etc.), with those of a loss of behavior disorder, impulse control, and even with features of obsessive-compulsive disorder (OCD) [21].

In addition, there is a difficult compensation problem: the game is basic to the evolution of a person, from the simplest ludic aspect to social development. How can healthy behavior become unhealthy [5]?

At what point does a recreational activity for adults, that entails positive consequences, such as the promotion of creativity, the development of psychomotricity, language, social relationships, pleasure, etc., incorporate elements such as chance and the possibility of betting? Clearly, at what point does it become a type of entertainment that involves risks? [24].

### 4.1. Limitations

The methodology can be considered, in some cases, a limitation of the study, not because of its validity or the relevance of the information but for the absence of experimental methodology in the study and the use of a particular adaptation of PRISMA methodology; more than this, the paper has been written with a clear intersubjective consideration, so the aim of an objective perspective of gambling consequences was not a must during the development on the article. In any case, the controversy among researchers continues that it is part of a type of behavioral addiction and the kind of necessary intervention for its recovery [25], while for others, it should be diagnosed as an impulse control disorder [7,20].

### 4.2. Conclusions

There is no global consensus about the measures, the causes, and the consequences for gambling implication. What is really clear is that we have accepted, integrated, and assumed the online bets and games into our lives, and that is a big success about the use of the OW in a situation that did not exist 20 years ago and is now a global problem with social, epidemiological, and clinical consequences. To limit the impact of pathological gambling, restrictive access legislation could be implemented, especially for minors. Access regulation could help reduce the costs and social problems of pathological gambling. However, for that, it seems that a reverse OW process would be needed. In light of the conflicts of interest related to the benefits of gambling, it is difficult actually to think of such an investment nowadays. A bitter victory, except for a few gambling operators.

## Figures and Tables

**Figure 1 ijerph-20-05481-f001:**
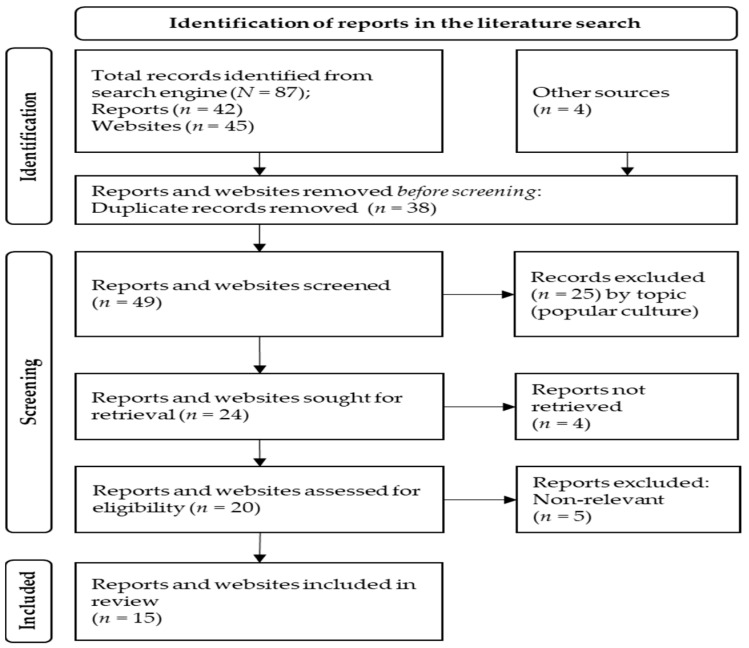
Flowchart for the literature review.

**Figure 2 ijerph-20-05481-f002:**
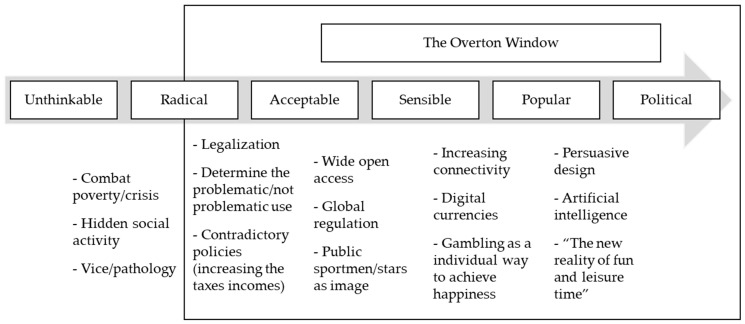
The Overton window in gambling regulation.

**Table 1 ijerph-20-05481-t001:** Overton window phases and verbatim examples.

From the Unthinkable to the Radical	“The situation led to gambling being legalized again, as for many this was the only prospect of alleviating the grinding poverty which they suffered through. Somehow seeking they wanted to recover the losses caused by the financial “crack”.”
From the Radical to the Acceptable	“When the internet started working, casinos too started operating online. Poker, slots, and bingo were a few of the real money games that were accessible to online gamblers. Online registration for games increased as they got better in terms of sound, graphics, and overall functionality.”
From the Acceptable to the Sensible	“The best online casinos have identified a market and have stepped up to deliver. More than this, immediately gambling operators linked their images to worldwide famous sportsmen and clubs, such as Real Madrid, Lionel Messi, Cristiano Ronaldo, Rafael Nadal, Neymar Jr…same time than other celebrities and stars were recognized as users of gambling online.”
From the Sensible to the Popular	“It has brought gambling closer than ever because the majority of online gambling platforms now allow connectivity with mobile devices.”
From the Popular to the Political	“The use of virtual reality technology is just beginning to be practical, and it is likely that the next phase of gambling will be related to VR: the opportunity to determine as a social activity with your own “friends”, in a real “room” shared with other people, and to receive all the stimulations without the risk and consequences of leaving home is really alluring and beneficial for gambler operators and involved stakeholders.”

## Data Availability

Not applicable.

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
