# Peer review of "Regulating Gambling Use through the Overton Window: From an Addictive Behavior to a Social and Epidemiological Problem"

_ijerph, 2023, doi:10.3390/ijerph20085481_

Round 1

Reviewer 1 Report

The article "Regulating gambling use through the Overton Window: from an addictive behavior to a social and epidemiological problem" aims to identify if an Overton Window strategy has been used to bias the adequacy of the gambling debate, as well as its scientific, legal and political bases and the main consequences for both the general population and the high-risk groups. A qualitative approach with a historical-logical method was used for this purpose.

Introduction

The introduction describes the background of the problem addressed, in terms of both legal aspects and the situational context of Spain.

In line 55, "Fenichel and other authors" should be adjusted, since it is only Fenichel in the first study and Cia in the second study mentioned.

Lines 64 to 66, do not mention Gaming disorder considered in the ICD-11 and included in the DSM-5 as an entity that requires more research, however it is important to mention it considering the fact to be included in the behavioral addictions category and it is mentioned again in the discussion including the proposed link between these two entities.

Line 66 "DSM-V" should be modified to DSM-5 and "pathological gambling" should be replaced by "Gambling Disorder" as a suggested name, in order to avoid issues concerning stigma and other related topics.

Lines 84-138, consider adding a figure showing this information in order to improve clarity and comprehension of this information.

Materials and methods

This section needs to be reorganized since the explanation of the historical research and the hermeneutic method can be synthesized and reordered.

The objective stated in the abstract and the objective in this section vary, it is necessary to harmonize them.

In the data analysis, I suggest including a flow chart according from the search algorithm of the included and excluded articles and finally including those selected for the analysis.

Results

In lines 190-193, content is duplicated.

Evaluate the usefulness of the suggested figure for the text in lines 84-138, adding the findings by phase and summarizing the text included in each stage.

Discussion

Lines 305-313, with missing references.

Similar to the introduction, replace "DSM-V" with DSM-5 throughout this section.

Expand the discussion according the proposed theoretical framework and background.

·        Consider a more detailed explanation and discussion of the link and implications of the behavioral addictions mentioned in lines 338-343.

·        Similarly, a deeper discussion of the reasons for combining/ mixing elements of addictive disorders with other Psychiatric disorders and explaining that the symptoms may not be due to a spectrum which includes solely substance use disorders like other authors and classifications proposed.

·        The implications of the aforementioned legalization and regulation policies on the specific population should be explored in more detail if possible.

Concerning the limitations, I consider that the fact of conducting a qualitative study with no experiments shouldn't be seen as a limitation due the type of research proposed.

Author Response

RESPONSES TO REVISOR 1

The article "Regulating gambling use through the Overton Window: from an addictive behavior to a social and epidemiological problem" aims to identify if an Overton Window strategy has been used to bias the adequacy of the gambling debate, as well as its scientific, legal and political bases and the main consequences for both the general population and the high-risk groups. A qualitative approach with a historical-logical method was used for this purpose.

Introduction

The introduction describes the background of the problem addressed, in terms of both legal aspects and the situational context of Spain. Thank you very much for your comment.

In line 55, "Fenichel and other authors" should be adjusted, since it is only Fenichel in the first study and Cia in the second study mentioned. Thank you for your appreciation, the change has been done.

Lines 64 to 66, do not mention Gaming disorder considered in the ICD-11 and included in the DSM-5 as an entity that requires more research, however it is important to mention it considering the fact to be included in the behavioral addictions category and it is mentioned again in the discussion including the proposed link between these two entities. Thank you for your comment, we have included the concept and the link in both parts of the paper.

Line 66 "DSM-V" should be modified to DSM-5 and "pathological gambling" should be replaced by "Gambling Disorder" as a suggested name, in order to avoid issues concerning stigma and other related topics. Thank you for your comment, the idea was not to repeat continuosly “gambling disorder”, but it’s clear the suggestion.

Lines 84-138, consider adding a figure showing this information in order to improve clarity and comprehension of this information. Lot of thanks for your comment, we have added a figure to clarify the information

Materials and methods

This section needs to be reorganized since the explanation of the historical research and the hermeneutic method can be synthesized and reordered. Thanks a lot for your comment and proposals, we have done a global revission of methodology part and we have included more information using the requirements of the reviewer, about the analysys, the framework and the data. We hope it would be now more detailed and with deeper and with wider implications.

.

The objective stated in the abstract and the objective in this section vary, it is necessary to harmonize them. Lot of thanks for your comment, we have harmonized the objective.

In the data analysis, I suggest including a flow chart according from the search algorithm of the included and excluded articles and finally including those selected for the analysis. Thanks a lot for your comment and proposals, we have included the flow chart, the theoretical basis and the selection we have done.

Results

In lines 190-193, content is duplicated. Thank you for your appreciation, we have deleted the duplicated content.

Evaluate the usefulness of the suggested figure for the text in lines 84-138, adding the findings by phase and summarizing the text included in each stage. Thanks a lot for your comment and proposals, we have done a table summarizing most signficitativa verbatim about the topic and a figure including main topics.

Discussion

Lines 305-313, with missing references. Thank you for your comment, we have included the references.

Similar to the introduction, replace "DSM-V" with DSM-5 throughout this section. Done again, thank you.

Expand the discussion according the proposed theoretical framework and background.

· Consider a more detailed explanation and discussion of the link and implications of the behavioral addictions mentioned in lines 338-343.

· Similarly, a deeper discussion of the reasons for combining/ mixing elements of addictive disorders with other Psychiatric disorders and explaining that the symptoms may not be due to a spectrum which includes solely substance use disorders like other authors and classifications proposed.

· The implications of the aforementioned legalization and regulation policies on the specific population should be explored in more detail if possible.

Thank you very much for your proposal, we have adapted the discussion to the requirements of the reviewer, we hope it would be now more detailed and with deeper and with wider implications.

Concerning the limitations, I consider that the fact of conducting a qualitative study with no experiments shouldn't be seen as a limitation due the type of research proposed. Lot of thanks for your comment, we don´t consider it a full limitation, so we have moderated the sentence about it.

Reviewer 2 Report

The authors analyze whether the Overton window has been used to influence society about its opinion on the practice of gambling. They applied the logical-historical method and qualitative research content analysis. They identified as main consequences: the acceptance of gambling for economic reasons, the use of popular characters to increase the acceptance of the phenomenon, the inclusion of gambling operators as risk control agents, and the absence of intervention until it became an epidemiological problem. 

My comments are:

1.-It is suggested to include a linking paragraph before the paragraph where you begin to explain the Overton window.

2.-It is suggested to include/increase statistical data about how the problem studied affects society in order to reinforce its seriousness.

3.-The paragraph of lines 178-183 needs to be explained in greater detail.

4.-Lines 190-191 are repeated in lines 192-193.

5.-I consider it necessary to broaden the search and deepen the analysis of the results. Likewise, include more information on the topics that are of interest to researchers, according to what you define as your objectives. It is suggested to reinforce what has been written so far in each of the phases of the Overton window so that its relationship with the definition of each of them can be more clearly observed.

6.-It is suggested to include a diagram that summarizes the application of the Overton window in the topic studied, containing keywords, concepts, or sentences for each phase.

Author Response

ESPONSES TO REVISOR 2

The authors analyze whether the Overton window has been used to influence society about its opinion on the practice of gambling. They applied the logical-historical method and qualitative research content analysis. They identified as main consequences: the acceptance of gambling for economic reasons, the use of popular characters to increase the acceptance of the phenomenon, the inclusion of gambling operators as risk control agents, and the absence of intervention until it became an epidemiological problem.

My comments are:

1.-It is suggested to include a linking paragraph before the paragraph where you begin to explain the Overton window. Thank you very much for your suggestion, we have included a few lines before the Overton window’s paragraph to connect both topics.

2.-It is suggested to include/increase statistical data about how the problem studied affects society in order to reinforce its seriousness. Thank you very much for your suggestion, we have included a few lines in the previously mentioned paragraph including statistics about the relavance of the topic worldwide.

3.-The paragraph of lines 178-183 needs to be explained in greater detail. Lot of thanks for your comment, we have changed the lines and explained the search strategy and the methodology we have used. Also, we have included a line in limitations about the possible lacks in the method.

4.-Lines 190-191 are repeated in lines 192-193. Thank you for your comment, the repeated lines have been deleted.

5.-I consider it necessary to broaden the search and deepen the analysis of the results. Likewise, include more information on the topics that are of interest to researchers, according to what you define as your objectives. It is suggested to reinforce what has been written so far in each of the phases of the Overton window so that its relationship with the definition of each of them can be more clearly observed.Thank you very much for your proposal, we have included more information using the requirements of the reviewer, about the analysys, the framework and the data. We hope it would be now more detailed and with deeper and with wider implications.

6.-It is suggested to include a diagram that summarizes the application of the Overton window in the topic studied, containing keywords, concepts, or sentences for each phase. Lot of thanks for your comment, we have included figures and tables to explain the Overton Window, the search strategy and to summarize the application of the Overton Window.

Round 2

Reviewer 1 Report

The manuscript has been substantially improved. I recommend it is in satisfactory state for publication.